# Compliance with COVID-19 government guidance and rules by disabled people and people from minoritised ethnic groups: Qualitative findings from the CICADA study

**Carol Rivas** *, **Kusha Anand**, **Amanda P. Moore**

Social Research Institute, University College London, London, England, United Kingdom

* c.rivas@ucl.ac.uk

**Data Availability Statement:** The full data cannot be shared publicly because they include responses from undocumented migrants who have refused

## Abstract

Within the 2020/21 CICADA (Coronavirus Intersectionalities: Chronic Conditions or Disabilities and Migrants and other Ethnic minorities) study, we explored full, partial or noncompliance with government COVID-19 infection-containment measures by people from minoritised ethnic groups with a disabling health condition or impairment. We used an assets-based intersectional approach and purposive sampling, included non-disabled and White British comparators, and trained community co-researchers to help us reach undocumented migrants and asylum seekers. We undertook 271 semi-structured qualitative interviews, followed by participatory workshops with interviewees to explore data and changes in experiences five and 10 months after the interviews. Perceiving their vulnerability to COVID-19, most participants quickly and often zealously adopted infection-containment behaviours, and continued this after restrictions were lifted. This could reduce mental wellbeing, especially in community-facing cultures, and could create family conflict. Various structural inequities impeded compliance. Many, especially undocumented migrants, felt imprisoned. The intersection of gender, citizenship, socioeconomic status and culture impacted disclosures of COVID-19 infection, support seeking and use. Many were unclear what was safe as well as unsafe. People complained that disability and cultural considerations were omitted from policymaking. Participants mostly had taken the COVID-19 vaccine by October 2022, but ethnic minority participants needed time to deliberate and trusted, community-embedded information whereas White British participants were mostly influenced by mass media. The intersection of health condition or impairment, poverty, and living alone led to more non-compliance with general rules, and more vaccine hesitancy than did misinformation spread through ethnic community channels. Many participants were reluctant to reintegrate in May 2022 because of continued perceived vulnerability to COVID-19 but by September 2022 = seemed more concerned about the economic crisis. We add two new 'types' to existing compliance typologies: deliberators (who eventually decide to follow the rules), and 'necessity-driven non-compliers' who are totally unable to comply because of their disabilities.

this permission. The anonymised qualitative data from interviews and workshops will be deposited for archiving and reuse under a Restrictive Licence according to UCL protocols existing at the time. The Restriction is in place to protect the identities of participants who have multiple protected characteristics described in the data and to avoid anonymisation being breached. These data will be available on request for up to 25 years after end of the study (i.e. up to 2047). Archived data will be checked for anonymisation before sharing; raw data will never be shared but will remain in the UCL safe haven. Data that are considered by the custodian to be sensitive and not in the public interest will not be shared despite anonymisation. Other anonymised data will be shared according to a Restrictive Licence and extant UCL protocols. The custodian of the data to whom requests may be made is Professor Carol Rivas, c.rivas@ucl.ac.uk; requests may also be made via researchdatarepository@ucl.ac.uk.

**Funding:** This paper presents independent research commissioned by the National Institute for Health and Care Research (NIHR). The views and opinions expressed by authors in this publication are those of the authors and do not necessarily reflect those of the NHS, the NIHR, MRC, CCF, NETSCC, the NIHR HS&DR programme or the Department of Health. The views and opinions expressed by the interviewees in this publication are those of the interviewees and do not necessarily reflect those of the authors, those of the NHS, the NIHR, MRC, CCF, NETSCC, the NIHR HS&DR programme or the Department of Health. The funders had no role in study design, data collection and analysis, decision to publish, or preparation of the manuscript.

**Competing interests:** The authors have declared that no competing interests exist.

# Introduction

In May 2023 the World Health Organisation (WHO) downgraded COVID-19 from a public health emergency of international concern (PHEIC) to an established and ongoing health issue [1]. Its 2023–2025 COVID-19 Strategic Preparedness and Response Plan [2] aims to guide countries transitioning to the long-term management of COVID-19. Despite this, in the UK, public responses to COVID-19 remain mixed. Most people have resumed normal lifestyles, with the pandemic becoming a marker of lost time, life stood still, or the cause of a current decline in personal circumstance [3, 4]. Some continue to mask up in crowded situations, especially in winter, because they are immunocompromised [5], or simply want to safeguard their health against COVID-19, colds and influenza, perhaps for an important event. COVID-19 remains disruptive, with for example 344 COVID-19-registered deaths and 6,832 people testing positive in England in the penultimate week of October 2023 (with many others not testing) [6]. There are also fears for the future; one estimate suggests a 47–57% probability of an equally deadly pandemic within the next 25 years [7].

Given this context and the different responses to pandemic and post-pandemic infection it is instructive to consider perspectives on the way the pandemic was handled and how responses changed through the pandemic itself. This can inform inquiries such as the UK COVID-19 public inquiry ongoing at the time of writing [8], as well as policy and practice, and was explored in the CICADA (Coronavirus Intersectionalities: Chronic Conditions or Disabilities and Migrants and other Ethnic minorities) study. We focused on people from minoritised ethnic groups who had a health condition or impairment that led to reduced everyday functioning. This subsection of the population experienced particular issues during the pandemic and often, as our data show, was wrongly viewed negatively for a supposed lack of adoption of infection-containment measures.

# Background

It is now well established that some groups are particularly likely to die from COVID-19, for example people from some minoritised ethnic groups, those with chronic health conditions or impairments, older people, the less well off, and 'essential' workers [9–12]. In the Omicron variant wave of November 2021, Bangladeshi and Pakistani people were 2–3 times more likely to die from COVID-19 than White British people [13]. Similarly, disabled people constituted 16% of the population, but by July 2020 accounted for 59% of COVID-19 deaths [9]. Between July and December 2021, when we undertook over two-thirds of our interviews, 2.5 times more deaths occurred in the most deprived than the least deprived areas of England [13]. People living in areas of multiple deprivation are more likely to be from disabled and minoritised ethnic groups. People aged 50–69 years old in the 10% most-deprived areas have double the likelihood of having two or more long-term conditions compared with those in the 10% least-deprived areas [13].

Structural inequities create barriers to the uptake of COVID-19 protection measures, for example for people in overcrowded accommodation, without savings (financial reserves to fall back on), on zero-hour contracts, or who cannot work from home, and are thought to part-explain inequitable COVID mortality statistics [9–20]. Low education levels, poor English language fluency, health illiteracy and a lack of appropriate and culturally targeted health information, are further symptoms of structural inequities limiting uptake among minoritised ethnic populations and those with disabilities and chronic conditions [21–28]. The impacts on vaccine hesitancy have been a particular focus of the UK government and mass media [29]; several surveys in 2021 and 2022 indicated significantly higher rates amongst some UK ethnic

minorities [30, 31]. Decisions around mask-wearing have been linked to explicit racism, with masks thought to exacerbate racial profiling [32].

Low uptake of COVID-19 protection measures among some minoritised ethnic groups occurred despite their relatively great sense of collectivist responsibility [15, 33]. For example, Atchison et al [17] surveyed 2,108 UK-resident adults in March 2020, with an 84.3% response rate. They found ability to self-isolate was lower in so-called black and minority ethnic groups but willingness to do so was not.

While there has been some attempt to understand the issues among minoritised ethnic groups [34, 35], there has been less attention paid to people with disabilities or chronic conditions, undocumented migrants, or to the intersection of ethnicity, disability and residency status. This was the focus of CICADA.

## Methods

CICADA was a mixed methods longitudinal UK-based study undertaken from May 2021 to October 2022, including a rapid review and systematic reviews, secondary cohort analyses, a new three-wave survey, three waves of qualitative data generation, and participatory and collaborative approaches to data collection and output development. Its aim was to contribute and inform evidence-based formal and informal strategies, guidelines, recommendations and interventions for health and social care policy and practice, to mitigate inequities and improve the experiences and social, health and wellbeing outcomes of minoritised ethnic groups at the intersection with disabling chronic conditions or impairments. The study was funded by the UK National Institute for Health and Care Research (NIHR). The published protocol [36] details the methods used.

Here we report only on our qualitative interviews and follow-on workshops with members of the public. We undertook 210 semi-structured interviews between 1st July and 20th October 2021, and 67 in 2022 (1st May-15th September) following top-up funding, totalling 271 interviews. Through purposive quota sampling, we recruited people from minoritised ethnic groups, and with diverse chronic conditions and impairments that led to their being disabled in their daily lives (a distinction used in line with the social model of disability [37]). We included all disabilities. Our ethnicity focus was people who self-identified as Arab, Central/East European, African or South Asian (see Table 1), to reflect recent migration waves and capture the perspectives of those at most risk from COVID-19. We targeted and grouped data according to six sites in England (London, Yorkshire, the Northwest, Northeast, Southeast and Midlands) for diversity in dimensions such as rurality/urbanity, deprivation level, service concentration and other factors relevant to our enquiry. We also sampled some native White British and non-disabled comparators (Table 1), congruent with our planned trans-categorical analysis, which avoids reifying or othering different ethnic minoritisations and enables consideration of the impact of structural forms of oppression across participants [38].

Full Institute of Education, University College London, Research Ethics Approval (UCL IoE REC 1450 Covid-19) for this study was obtained on 21st July 2020, before the study commenced and an amendment approved 30 July 2021, to enable us to record co-researcher training (with consent), co-researcher face to face convenience sampling and recruitment from their immediate networks, co-researchers to use local charity premises for interviews, and oral (audiorecorded) consent where an undocumented migrant did not want to write their name. Subsequently Health Research Authority (HRA) approval (IRAS project ID: 310741 Protocol number: NIHR132914 REC reference: 22/SW/0002, Sponsor UCL/UCLH Joint Research Office) was obtained on 18th February 2022 to enable a change in recruitment to our separate survey in the final months of the study. All participants provided full informed consent; mostly

**Table 1. Interview participant demographics (n = 271).** Figures add up to more than 100% due to rounding, and for conditions because some participants had comorbidities. We estimate there were approximately 40 undocumented migrant respondents but most asked not to be identified as such in our data even with anonymisation.

| | | |
|---|---|---|
| Ethnicity | South Asian (SA) | 35.42% |
| | North African (NA) | 17.71% |
| | Other Arab League (AL) | 8.86% |
| | Central/East European (CE) | 9.96% |
| | African (AF) | 11.07% |
| | Undocumented (UD) | 2.21% |
| | Native White—British, Irish (WB) | 7.38% |
| | Caribbean (C) | 0.74% |
| | Southern European (SE) | 3.32% |
| | Mixed race (MR) | 3.32% |
| Age | 18–24 | 11.44% |
| | 25–34 | 36.90% |
| | 35–44 | 18.82% |
| | 45–54 | 11.44% |
| | 55–64 | 5.54% |
| | 65–74 | 2.216% |
| | 75+ | 0.37% |
| | Unknown | 12.55% |
| Gender | Male | 46.49% |
| | Female | 51.66% |
| | Gender non-conforming | 1.11% |
| Site | South England | 7.75% |
| | London | 39.85% |
| | Midlands | 11.07% |
| | Manchester and NW Coast | 13.28% |
| | Yorkshire | 11.81% |
| | Cumbria and Newcastle area | 6.27% |
| | Scotland, Wales | 8.86% |
| | Unknown | 0.37% |
| Condition | | N% alone |
| | | (N% Including with other conditions) |
| | Dexterity | 1.12% (1.38%) |
| | Mental health | 7.01% (31.75%) |
| | Mobility | 12.92% (35.59% |
| | Stamina/breathing/ fatigue (incl. heart, lung) | 16.24% (43.88%) |
| | Sensorial | 1.85% (6.48%) |
| | Cognitive | 1.48% (11.20%) |
| | Food-relevant | 6.64% (22.65%) |
| | Brain hyperexcitability (migraines, epilepsy) | 2.58% (3.44%) |
| | Cancer | 5.17% (6.77%) |
| | No condition/Impairment (across ethnic groups) | 9.59% |
| | 2+ conditions | 33.58% |

this was written (with some forms copied and emailed to us) but a few, as per our ethics revision, preferred distorted voice verbal consent, which was recorded. Five undocumented migrants used false names to preserve anonymity. Most consent was recorded by core team members, with 32 taken by a partner charity and 48 by lay trained co-researchers, who

discussed the process with us in debriefs. After analysing interviews, we found some participants did not fit our initial criteria despite pre-interview screening. For example, participants who self-identify as South Asian but have one White British parent would likely have different experiences to someone with two South Asian parents. We note, in keeping with intersectionality theory, that all categorisations are inherently problematic (which we discuss in a different paper [39]). Our sampling was designed to accommodate practicality, basic expectations of rigor in single-axis (single factor) reporting [40] and inter-categorical, intra-categorical and trans-categorical analysis [38]. For in-depth perspectives we only invited to follow-on discussions in two series of workshops the 134 interviewees from 2021 who tightly satisfied our initial sampling criteria and for whom we had contact details. In May 2022, approximately five months after interviews, 104 attended to explore changes in their experiences and validate our interpretations of findings; 35 attended workshops approximately 10 months after the interviews. Workshops used ideation tools such as journey mapping and structured brainstorming [41] to provide structure and support contributions.

The analysis reported here used a deductive-inductive framework approach [42] on our combined interview and workshop datasets. Deductive themes, developed from our research questions, encompassed extant experiences and recollections back to January 2020 and explored all aspects of daily life and formal and informal networks. A lead researcher independently coded different triplets of transcripts in pairs with four junior researchers (partly due to our commitment to develop early career researchers and partly because of dataset size). Coding was compared and the process repeated until good concordance (>75%) was reached across the data. The remaining transcripts were divided by ethnic group with each researcher responsible for coding one or two groups (including 'all others'); the study lead (CR) randomly quality-checked samples. Inductive themes were identified and refined through discussion. Data were summarised in Excel charts for each topic and key extracts copied verbatim. Charts helped identify common patterns within and between ethnicity and disability categorisations including White British and non-disabled comparators; explanations were sought for divergence. A sensitivity analysis considered the impact on findings of including interviews not fitting our core inclusion criteria (see S1 Table).

The data reported here describe participant perspectives and experiences regarding government responses to COVID-19 and the main infection-containment measures, to support changes in health and social care; other data and forms of analyses are reported in other papers.

## Findings

Interviews lasted 25–90 minutes, workshops approximately 2 hours. Participant details are shown in Table 1. Considering both ethnicity-disability combinations and single categories, these appear broadly representative of national data [43, 44] and show good diversity in line with qualitative approaches. All included ethnicity-disability intersectional axes were sampled except African or Central/East European participants with sensory loss. South Asians form our largest ethnic dataset and Central/East Europeans our smallest (though we aimed for equal representation across groups). More participants lived in London than elsewhere, reflecting national migrant distributions, and more were aged under 55, possibly reflecting pandemic restrictions and our predominant use of remote interviewing. Females slightly outnumbered males.

To better contextualise their experiences, we have embedded them within the changing national pandemic situation over the 34 months (January 2020-October 2022) they cover.

## The pandemic starts

SARS-CoV-2 (COVID-19) emerged in China between early October and mid-November 2019 [45] and by January 2020 had spread globally, with the WHO categorising COVID-19 as a public health emergency of international concern on 30 January 2020 [46]. The first officially confirmed UK case was reported on 31 January 2020 [45]. However, anecdotal evidence indicates several unconfirmed UK cases in early- to mid-January 2020 [47]. The first European case retrospectively confirmed from genetic and epidemiological studies occurred in December 2019 (official figures state January 2020) [45]. On 11 March 2020, the WHO declared a pandemic [48]. This had been expected since February [49], thus on 3 March 2020, the UK government pre-emptively published its Coronavirus action plan [50]. Initially, this focused on increasing health and care system capacity to manage COVID-19 illness. From 16 March 2020, the public was encouraged to avoid non-essential contact with others, work from home where possible and avoid hospitality, social venues or mass gatherings [51]. This became legally enforceable from 26 March 2020 after the first UK-wide lockdown began on 23 March 2020, as COVID-19 spread [51, 52]. The lockdown (phased out in May 2020) required non-essential businesses and venues to close, and the population to stay home except to: shop for necessities, take one form of exercise a day, get medical or social care, or commute to work if necessary [51, 52]. Public gatherings of more than two people were prohibited and face-to-face teaching in schools and universities stopped, with some exceptions [51, 52]. Though arguably necessary, there is considerable evidence, which we add to, that lockdowns and shielding affected people's mental health and wellbeing [53]. At the time, a few CICADA participants across ethnic groups reflected that they had doubted the pandemic was genuine or concerning (*"At first I thought it was a political move from the government to gain profits or to control people."* P25,NA) and others thought it was overhyped (*"I think in the beginning it really got blown up [by the government and news]".* P277,CE).

A few others had considered it a punishment from God because of sins. A rumour described by three participants from Pakistan framed it as a plot to kill people from minoritised ethnic groups. Misinformation (wrong information shared non-maliciously), disinformation (information manipulated or fabricated to harm) and malinformation (genuine information used to inflict harm) was rife in social media and other outlets [54–56]. Despite the absence of rumours from our White British data, other studies provide only weak evidence for their greater concentration, uptake and sharing among minoritised ethnic groups in the UK [57, 58].

Mostly participants were frightened by alarmist talk in the news and social media (*"The news wasn't helping at all because it was just death after death after death. [. . .] it made me feel scared of people."* P199,SA), followed the guidelines carefully, even over-zealously, and sometimes talked about the world as they knew it coming to an end. This was especially remarked on by participants in the Midlands, Southeast and the Devolved Nations.

*I have created a springboard of changes to protect myself and my family members. All the government guidelines make it to the springboard. I've created a virtual chart of rules to visit me and meet my kids. I know this is offensive for my family members and community. To reduce transmission, we are following the total digital triage, redesigning physical spaces, and keeping emergency care handy for my family members. I've also changed my working style, following remote triage, and accessing multi-modal remote consultation via telephone, video, and online.*

(P70,SA)

Almost all African, Arab League, Undocumented, Mixed Race and Caribbean participants complied carefully. Undocumented migrants had a specific motivation to comply carefully, worried that infection could lead to deportation ("*My friends are also scared because of their immigration status. They were wearing triple masks.*" P103,UD). By contrast, only a third of people from Central/East Europe, half of White British and two-thirds of European Union (EU) comparators adhered to the guidelines. One White British person specified breaking the rules conscientiously, highlighting the importance of agency in non-compliance decisions:

> *[I used] the government guidelines but also using my judgement because I do not trust the judgement of some people, but I do trust my own judgement. I would never break the rules to put anyone else at risk, but I couldn't see the point of staying close to where you live [. . .] In this country all the rules are entirely voluntary. You rely on people to do the right thing and they rely on you to do the right thing. Police enforcement was minimal. Only a few 1000 people got fined and that was minimal. Maybe that was a good thing. Because I do not like to think of constraint, because anything I do, I'm imposing on myself. No one was making me do things, I'm doing it because I want to and a vast majority of people I think thought the same way.*

> (P238,WB)

Critics across the minoritised ethnic groups said the government designed policies for the mainstream population without considering the considerable constraints many immigrants had in complying because of their living conditions, their greater likelihood of being key workers, or the costs involved and their lack of recourse to government support or private transport ("*and I don't have a car. [. . .] they just completely overlooked the fact that not everyone has access to transport*". P73, MR). This intersection with socioeconomic status led several to only partially comply (e.g. washing hands but not using masks). Two Caribbean participants commented that class or socioeconomic status was a greater divide than ethnicity or disability.

> *Wearing a mask is quite mandatory but also it comes as a challenge since the masks we wear we need to change them once in a while right. Now obtaining them it's another cost also.*

> (P52,NA)

> *So I had to accept the mortality of it all, we both did. And we just knew that if the worst happened then it would happen. I think we couldn't worry about it anymore, we just had to accept it. So that took away a lot of my guilt about leaving every day [as a key worker] because I was given that guilt that you are going to kill me [her husband]. But I did have to accept that I had to just go out.*

> (P204,CE)

Participants with upper limb loss, autism, sensorial loss or in wheelchairs also found compliance challenging across ethnic groups and felt omitted from the guidance, leaving a few totally unable to comply.

> *I'm single-handed. I do wash my hand, but you know when you get [in the] workplace, people expect you to wash your hands like other people wash hands but the company doesn't have the equipment for people like me. So, it becomes quite a problem.*

> (P90,SA)

*The biggest issue for somebody visually impaired, we rely on touch. So with this pandemic, obviously, if you're going out, you can't touch anywhere you have to be with face mask, gloves, there's so many things, then obviously, it could be dangerous with visual impairment, when you can't see even like a two-metre distance. How can we see a two-metre distance if we can't see? It's very important because if I have to go on my own, even for just a walk, how do I get anywhere? There are many places I have to touch [. . .] a door or gate [. . .] they have the numbers to type and enter. That's one of the things and when you go out and about, you touch. It's a bus, or a bus stop, or something we just have to touch. We rely on touch. It means everywhere you have to sanitise, and you can't sanitise every second.*

(P5,SA)

## Attitudes towards the government and information provision and access

Overall, as the pandemic developed, most participants saw it as a public health crisis, which led to reflection on government responses. Before the discoveries leading to the 2023 COVID-19 enquiry [8], more people were positive about the public health response than not. Some said the Government was trying to protect people which helped them stay confident–a view most common in South Asian, African, Arab League and Undocumented participants, the Devolved Nations and Southeast England. Those with sensorial impairments or cancer were most likely to hold polarising views either for or against the government at this time. Overall, our data showed the majority of those expressing an opinion of the government ended up critical of it. In particular, the Central/East European, non-core ethnic groups and South Asian participants, and those with multi-morbidities or mobility issues made 3–6 times as many anti- as pro-government comments (counting all comments in the data, whichever pandemic stage they referred to), while the White British participants contributed only criticisms. Critics said the Government (*"these amateurs"*) waited too long to act, did not respond to scientific advice, had inconsistent messaging, and eventually also that it broke its own rules:

*You hear the government saying, "You don't need to do this and you won't be fined. But we recommend you do something," and people just automatically assume you don't have to do anything and the pandemic is over.*

(P106,CE)

*They waited till all those people died in hospital and cases got worser that the hospital could not manage. The government don't \*&£!\* care, they're just bothered about lining their own pockets. Look at all the Covid contracts gave to friends and relatives. Even when we were in lockdown, the politicians were doing as they pleased, yet expect us to abide by rules.*

(P194,SA)

Participants also said Government information was unclear about what was safe, focusing rather on what was unsafe, which led to confusion among some minoritised ethnic communities, whose information preferences were un-met. They simply did not know what they **could** do. Participants further criticised information support for disabled people and information access disparities (for example concerning sensorial impairments, languages, generational differences and digital literacy and access).

*And also like when they were sending in all these letters saying we are high risk and we shouldn't be leaving the house maybe send in some more encouragement with it, like see how*

*we can help ourselves when we are isolating, instead of saying "you can just die if you leave the house, you can die if you travel, you can die if you sit in a bus". What are the alternatives for diabetics? So I do think they could have approached it differently [. . .] The fact that you just put in a letter in a pamphlet and just send it out is convenient but it doesn't work, and that's exactly what they're doing with vaccinations. Also, they send you an email or text message saying go and get yourself vaccinated. It doesn't help people who need something more relatable. If South Asian people aren't coming forward or aren't giving you a good uptake for vaccination, try and find out why and maybe also make a page with them, which is more relatable to them?*

(P166,SA)

*It was not clear what is more important, shielding as a diabetic patient who is African, or going out for shopping, food and exercising. The government advice was not clear for me, so I was not going out even for one hour to walk. I was very worried to get the virus. [. . .] To remove the uncertainties we have, the information is not enough particularly to ethnic groups.*

(P157,AF)

Accounts of good practice were rare and mostly came from initiatives within communities:

*We watched the news, Asian channels and on there the NHS told us about how to recognise the symptoms, what it was and how to keep safe. Our social worker explained as well in house meetings in Urdu as well as bringing us face masks and soap to wash our hands.*

(P177,SA)

The broader population, that is including people outside our study, also expressed mistrust in the government, causing failures in mobilising public COVID-containment behaviours more generally, and "solution aversion" [59]. This, and the mortality statistics showing the greater impact of COVID-19 infection on minoritised ethnic groups and disabled people, increased participant anxieties.

## Shielding, disability and access to resources

In March 2020, the Government compiled a list of people 'Clinically Extremely Vulnerable' (CEV) to the virus, an especially relevant policy for our participants. CEV people were advised to stay home (i.e. 'shield') for 12 weeks [60], which participants on the list said increased their anxiety. To ensure their continued access to food, the Government gave the CEV list to supermarkets to prioritise online slots for them, and established a national service of free, weekly, standardised food box deliveries via national food distributors, local authorities, and local voluntary groups. Food banks sometimes ran dry as demand for them increased, and their usage became capped, leading one person to describe being helped to move from food bank to food bank: *They gave us twelve times [the cap]. And whenever this one finished they gave us another address and I can go there as well.* (P226,SA)

Many participants were however not deemed eligible for the CEV list, despite pandemic control measures making shopping more problematic for them than for non-disabled people. A very few were in the process of diagnosis when the pandemic struck and so ought to have been on it but weren't. Many suffered from the impact of queues and long waits caused by social distancing and constraints on the numbers allowed in a shop at one time, especially those whose condition meant they could not stand for long, needed mobility aids, or found mask-wearing difficult.

*I couldn't wear a mask when out because of my COPD and anxiety so I made sure I either went out early morning or late at night when shops were less busy.*

(P190,SA)

Access to culturally- or faith-appropriate food also became problematic and intersected with health condition. Public order measures unnerved minoritised ethnic groups, especially but not only the Undocumented participants.

*But obviously most of the time the options were limited to cook our traditional food. My mum was reluctant to eat. I was worried about her diabetes and her weight.*

(P68,NA)

*The first time that I went to a food shop, it was the time when there was the rule of one person per household. There was police in front of the supermarket just to make sure that the queues were okay. I spent the next 20 minutes talking myself out of a panic attack because there was police there and I didn't know what to do, and I didn't know whether or not I was actually going to get told off, whether I was going to get arrested because they were just arresting people left, right and centre, because no one was sure what they were actually meant to be doing. It was chaotic.*

(P276,CE)

A few participants used the supermarket early morning openings dedicated to disabled people, older people and key workers. However, people could not use these if they needed support to get up, washed and dressed, or medical technologies such as feeding tubes or oxygen tanks, or assistance with mobility or transport, or needed manipulation therapy for musculoskeletal problems before they could move, or had assistance or guide dogs. All of these lengthened the time needed to leave their home and made them dependent on the timekeeping of those assisting them. Similar findings were reported by the charity Scope [61].

*I was waking up at the crack of dawn at like literally 5:00 AM in the morning when TESCO would open so that there's no one there. And I'd wear like double masks and gloves. I remember and be like extra extra cautious and just go in and go out.*

(P166, SA).

Issues such as this meant the CEV list was criticised by disabled people and the charities supporting them, eventually leading to its revision and expansion in February 2021, shortly before the CICADA study began [62].

## Mental health and wellbeing challenges of shielding and social distancing

Participants often shielded or kept away from friends, neighbours, or family, even when not on the CEV list, fearing their potentially greater vulnerability to COVID-19. Resultant loneliness was stated by just under a fifth of all participants, and left them bored ("*I sleep, I eat and sleep I eat*", *P94*, AF) and inward-looking which led to mental health issues.

*I was alone in the house. [. . .] Mentally, it had a great impact. There was that depression, I have never heard of the disease.*

(P145,AL)

*I'd be restless, angry, shouting [. . .] I'd be walking down the street and shouting all the way up and down. And people, people heard the shouting. . . . So they were saying, where is that noise coming from? I would say: I don't know. But it was me. . . Yeah. It was like something out the Exorcist.*

(P3,SA)

Members of the ethnic groups most likely to specify loneliness (a third of Arab League participants, and slightly fewer C/E European and African participants) tended to live in areas with low densities of people of similar heritage. Despite South Asian participants explaining they strongly felt the loss of their previously frequent community interactions, they did not as a rule feel lonely because they lived with several others, or connected well online or interacted in smaller groups. The loneliest by disability category were those with sensorial impairments (almost half) followed by one-fifth to one-third of those with prior mental health and cancer issues, neurodivergence and multi-morbidities; this goes against common misconceptions that all neurodivergent people and all people with mental health issues benefitted from the reduced socialisation of the pandemic. Very few of those with no disabilities felt lonely.

Loneliness led one participant to binge eat and seven to increase alcohol consumption, of whom four had prior mental health issues. Similar findings were reported in an international cross-sectional survey [63] and a major UK longitudinal survey [64], though a US three-wave survey found loneliness in people living alone or with disabilities was low before, and did not increase during, the pandemic [65].

Eleven participants across ethnic groups (including three White British) said they had felt suicidal because of the pandemic, seven of whom had prior mental health issues, with six living in London,

Shielding could also create stress within households, with discord and withdrawal from the family, when over-crowding caused tensions and a lack of privacy, or family members felt burdened from protecting the shielding person. This was described as particularly challenging within several multigenerational households, typically, but not only, South Asian. However, in several other cases it strengthened social ties between family members and friends, albeit that when undertaken remotely this could not replace in-person socialising.

*You're isolating with the people you're living with as well because you don't know where they've been or they've come back and they've probably got it and just because it was told it's a disease where, a condition where you can't see it or sometimes people don't have an effect for a few days and then it, you know, you can catch it or you can pass it on. All that, it was just, it was daunting to be honest with you.*

(P188,SA)

*I stayed mostly in my room from the beginning as I have teenage grandchildren who don't listen, keep going in and out the house. You know what boys are like they have their own mind! My son said it was causing arguments and I'd be safer staying in my room.*

(P167,SA)

Those from community-facing cultures, most commonly South Asian and African participants, noted social isolation was culturally problematic.

*I think that is cultural for sure, because I came from a family where there was always life all around, always people, even workers coming in and out, carpenters or somebody or the other,*

*my dad has always had people around fixing things, friends coming over, just dropping in for lunch without announcing, growing up like that.*

(P16,SA)

Local and global social distancing kept participants from faith, community and social gatherings and physical activities and physically separated from friends and family. Separation was often felt most acutely when intersecting with disability or when families lived in a participant's country of origin. This was also noted by Kulpińska, Górska and Wyrwisz [66].

*Some cultures are more connected to their communities. . . my parents normally would come, which is linked to ethnicity or culture, for a couple of months in the summer every year, so they didn't do that, so that was different, and normally I think in another culture, in a western culture parents don't come for that long normally when they come visiting, so yes, so that was a loss for us not to have them home.*

(P16,SA)

*We [Nigerians] are quite sociable people. We like gathering together in our churches, in our other religious settings and we party a lot as well. So, I used to go out a lot despite using a wheelchair, I used to go out with my friends. Also, they used to come over you know, I've had to obviously restrict that.*

(P2,AF)

Migration can reduce local social networks if the person has not moved into a diasporic community, making remote connections more important, more likely for our Central/East European, least likely for our South Asian participants.

*When government said, okay now you can have the bubbles, I didn't really have that option because I didn't know anyone there and I couldn't really have any bubble.*

(P74,CE)

## Contact tracing and patrols

To control movement, in addition to the lockdown and recommendations for shielding and social distancing, contact tracing was prominent in all four UK nations during May and June 2020 [51]. People were expected to leave contact details at all public spaces visited; this was temporarily mandated for track and trace (contact tracing), then became voluntary. QR codes appeared everywhere for the purpose, linked to a special government smartphone app. Many participants from minoritised ethnic groups were reluctant to comply, arguing a lack of clarity regarding Government use of the data, irrespective of citizenship or socio-economic status, though undocumented migrants were the most afraid, and so became virtual prisoners in their homes.

*The public authority was just assisting individuals with the right documentation. The greatest change I did was stay inside my home like a detainee [. . ..] my friends are like me terrified to reveal their movements due to their visa status.*

(P78,UD)

*The issues people of colour have with police and accessing services, I have not had to deal with [before the pandemic]. So it comes from the privilege an upper middle-class background . . . . [After Covid] there have been times when police had stopped. . .an amount of terror. And because you are Brown, and you hear the stories . . .even if you have not faced it before, the fear still exists. . . . When you had to give your details at restaurants or cafes, it felt very much like surveillance. As someone who doesn't trust the authorities, because of where I come from, I did not believe the details would just be used for track and trace.*

(P67,SA)

Similar findings were reported in a qualitative US study of 20 undocumented migrants [67]. Police, army and COVID wardens patrolled [68], looking for people breaking lockdowns, particularly worrying for minoritised ethnic groups as our data show. The situation was worsened by government official Dominic Cumming's breaking of the rules, with participants calling for his legal accountability.

## The easing of restrictions

As COVID-19 cases dipped through May and early June 2020, restrictions were eased in all four UK nations. Caps on the frequency of outdoor exercise were removed; social contact outdoors was allowed with a limited number of other households and schools began re-opening. By the end of June 2020, non-essential shops and other businesses could re-open. Northern Ireland was the first UK nation to allow indoor social contact, in groups of six, from 23 June 2020.

From June to September 2020, the UK experienced low rates of COVID-19 hospitalisation [69]. Pandemic fatigue, detrimental impacts from lockdowns [53], and heatwave summer weather, thought to lower viral transmission, reduced concern by the public and government alike. There was large-scale public defiance of pandemic rules [70]. Workers were encouraged to stop homeworking, which made many participants, vulnerable to COVID-19, fear contracting it from others. The UK Government turned to promoting economic growth, with Sunak's Eat Out to Help Out scheme encouraging social mixing to benefit the hospitality sector and wider economy [71]. The scheme was criticised for facilitating a spike in cases [71–73], has continued to be viewed as 'stupid', for example in 2023 mass media [74], and has been at the centre of the enquiry into the government's handling of the COVID-19 pandemic [74]. Public messaging became confusing for participants [75]. This was layered onto the emerging stories of members of parliament breaking the rules.

*The government speeches and messages were confusing for people like me having a condition. The political speeches and political leaders were misleading. This situation upset me as the political leaders are enjoying their lives. It feels like a hoax. There are many people including the political leaders who are clearly not following the government guidelines.*

(P85,SA)

## Mask-wearing

Masks had been recommended for staff and patients in health and social care settings from the start of the pandemic, but initially not for the public as there was a shortage for healthcare staff. In March 2020, the NHS website noted little evidence of widespread benefit for members of the public [76], reflecting scientific uncertainty [77].

As restrictions eased, in June 2020 England and Scotland, followed in July by Northern Ireland and Wales, made it a legal requirement for face masks (including home-made) to be worn in public spaces–starting with public transport and then moving in England on 24 July and Scotland on 10 July to include other spaces such as supermarkets, restaurants, and performance venues. In Wales, mask-wearing was recommended from 9 June 2020, mandated on public transport on 27 July and mandated in other settings in mid-September 2020. Many participants, fearing COVID-19 infection, welcomed this, saying mask-wearing had become part of their normal routine though South Asian, Arab League, Undocumented and White British participants and those living in Newcastle in particular were concerned many people ignored the mandate.

*God will not help you when you are just lying down doing nothing. You have to first help yourself, and helping yourself means you have to wear a mask, you have to maintain distance, you have to wash your hands, you have to be careful. Now when you do that, then the rest you leave to God.*

(P283,MR)

*It was October last year [2020]. I went into Oxford and the bus was heaving. It was full of people, all the windows were shut, it was hot in there. It was steaming, you know. And no one had masks on at all like.*

(P239,WB)

Mask-wearing issues were common across ethnic groups, affecting roughly a quarter overall. though least likely for White British, Central/East European and African participants. Muslims lacked suitable masks to wear with headscarves, especially noted by those from Yorkshire.

*And the other bit which is related to my culture and my religion is that these masks are not designed for someone who wears a scarf. No one thought about us [. . .] With openings on both ends so, that you can, and a plastic square that goes in and you would attack the mask to it. Otherwise, if we are lucky enough, we'd find one of these masks because these would work, you see.*

(P19,AL)

Mask issues were similarly common across all but the food-relevant and sensorial disability groups. Hand or arm amputation or cerebral palsy with associated sialorrhea (excessive salivation), made mask-wearing impractical or impossible. These participants felt forced to withdraw from un-masked exposure to others, increasing their isolation.

*Since I have one hand, I first hang it on my first ear either the left or the right then using the hand I put it on the next ear but in places where it's open and the wind blows it might be a challenge because now if you put it on the left ear and the wind blows you see the mask will be blown off.*

(P52,NA).

Participants with mental health issues often said they forgot their masks. Some with breathing problems specified 'suffocating' and being 'unable to breathe'. A few said they discarded the masks when sweating in the hot summer of 2020.

## Experiences of COVID-19 infection and bereavement

There were many examples of whole families self-isolating due to infection, exacerbating food issues, isolation and other problems. Minoritised ethnicity participants were more likely than the White British comparators to experience close COVID-related deaths (partners, friends and parents), some in participants' countries of origin. Some spoke of 'key worker' friends and family dying in the UK.

The impact of these bereavements was compounded by being unable to attend the funerals, an important rite of passage in many cultures. Travel restrictions made funerals outside the UK inaccessible. In the UK, at one point in 2020 only close family members could attend funerals [78] and in 2021 only up to 30 people including funeral services staff [79]. A South Asian participant explained how phone calls are no substitute. For Muslims, as in many religions, community interaction and shared mourning are central to grief processes and continuing bonds with the person lost [80].

*So many people in our community (Baradari) passed away. We could not pay our respects to their family or do the burial prayer (Janazah). It was really hard to deal with because the Pakistani community supports each other when there is a death in the family.*

(P176,SA)

When the main family provider died, families with limited or no access to social support funds experienced severe financial hardship, so some main provider participants avoided telling family they had COVID to minimise anticipatory anxiety.

*What was hard was accepting that I had contracted the virus and telling it to my people [. . .] the fear of telling it to my people was worrying, knowing it will stress them. Maybe they think I'm going to die, [that] I'm going to leave them. I'm the sole provider of my family, so it was a little bit difficult telling them.*

(P145,NA)

A few other participants did not disclose infection because they feared unhelpful responses, for example because of myths about whether COVID was real. One said COVID highlighted gendering within her culture; her infection was dismissed because she was female. She received neither care nor support, being undocumented.

*The hard thing was to tell my family. They didn't really care about it. I have to do a lot to convince family members. This was upsetting but they hardly listen to ladies in my culture . . .. . .I prayed to God. . .after 40 days of praying, one day I felt I don't have an COVID. I prayed every day'* (Undocumented, but does not want to be identified as such, hence no P number here).

## The second wave

The second UK COVID-19 wave began with a sustained rise in cases from September 2020 [81]. Attempts were made to contain it with local hotspot lockdowns, to avoid disrupting nationwide economic and social recovery; the first began in Leicester on 29 June 2020 [82]. A system of tiers followed in England with areas categorised by COVID-19 prevalence. Those living in tier one areas could mix with up to six people from different households inside or

outside (the 'rule-of-six'), whereas those in tier four could meet with one person outdoors [83]. Sometimes people on the two sides of one street were in different tiers, adding to confusion and frustration [84]. Worrying for many, especially the South Asian people in our sample, was that they lived in dense diasporic communities, in deprived areas, in conditions preventing adherence. Some South Asian participants felt the government blamed them for the second wave.

> *There are a lot of people in my town who are not following the government guidelines and sanitising hands. You won't believe there are about thirty people living in the same house. This is a hot spot in Leicester. You must be reading in the newspaper. I am paranoid and sad to see this situation. The local leaders first should check the housing and immigration in this area [...] Unless the government are prepared in this country to tackle some of the issues rather than putting the blame on ethnic minority then nothing can be done.*

> (P200,SA)

On 2 December 2020, the second national lockdown ended in England and on 15 December 2020 Boris Johnson, the British prime minister at the time, declared the relaxation of other COVID-19 containment rules over Christmas. People made Christmas travel, hosting and party plans, then on 19 December he backtracked and mandated people across London and the Southeast to stay home from 21 December 2020 [85], a requirement many ignored or circumvented by leaving the region by 20 December to continue with their plans. The Government's own partying at this time is now notorious.

## The vaccination spring

In early 2021, with the country still in lockdown, the COVID-19 vaccine was slowly rolled out, with prioritisation by government-presumed risk [86]. Restrictions eased in all four nations between March and May 2021 due partly to confidence in the vaccine, with schools re-opening and limited outdoor social mixing permitted [87]. Spread of the COVID-19 Delta variant caused the third wave of the pandemic, officially commencing in May 2021 when the CICADA study started. Public re-adoption of restrictive behaviours remained limited, again partly due to confidence in the vaccine. However, the CEV group remained anxious.

Most participants reported having had the COVID vaccine, with the highest uptake among White British and White Other participants, those with mobility issues and non-disabled comparators; 35 had not been vaccinated at the time of interview (especially Undocumented and Central/East European participants and those with sensorial impairments, mental ill health or neurodivergence), but 20 of these were considering it. Nineteen core participants and one of mixed race sought alternative or additional protection from traditional home remedies. This was proportionately more common amongst African (a sixth) and undocumented (almost one-third) participants but was represented by a few across all core ethnic minority groups. Only one person interviewed after December 2021, from Central Europe, was adamant they would not take the vaccine; they had no disabilities or health conditions and felt the benefits were minimal. Many stated the vaccine gave them greater confidence and freedom, as 'a protective shield', with positive impacts on mental health.

The vaccination decision-making process, however, was complex (Fig 1); many participants across ethnicities and health conditions weighed up, debated and deliberated the options for some time. They described influences from NHS, private and community ('family') doctors, mass and social media, friends, family, the community, faith leaders, the Council and government. Decision-making was also affected by personal circumstances, social responsibilisation,

**Fig 1. Schematic illustrating the complexity of vaccine decision-making.**

unfamiliarity with vaccines, preferences for natural remedies (especially among African and Undocumented participants but relevant for a few across the ethnic groups), spiritual beliefs, and self-efficacy in disregarding social pressures and misinformation. Experience of COVID-19 infection in close others did not consistently affect decisions.

Several Central/East European and Mixed Race participants complained that personal choice had been removed. Twenty participants (most notably around a quarter of the Central/East Europeans) had to take the vaccine to travel or work, and a few participants across the minoritised ethnic groups said they took it so they could receive medical care or socialise without worrying about COVID-19.

Other negative comments made by a few people across ethnic and disability groups included fear the vaccines were developed too quickly, despite the vaccine having been available for many months by the time of the interviews. Fear of side-effects was reported by 44 participants across ethnic and disability groups as one of the most significant factors affecting vaccine uptake. Considering ethnicity, this fear was particularly common in those from Arab and North African groups and least common in the White British and White Other participants. Diabetics in the food-relevant category may ironically have been affected by reports that COVID-19 infection triggered diabetes. Fear of side effects was also described by one-sixth of participants not disabled by their condition who feared it would be worsened, and by over a quarter of those with no impairment or chronic health condition (Fig 2). There was an intersection between disability, living alone or in poverty and ethnicity, which was described by a few participants with diabetes, mobility issues, cancer or multimorbidity, who were worried about the financial and practical implications of side effects.

Central/East European participants–often those with mental health issues—gave multiple justifications for vaccine hesitancy whereas other groups focused on one or two reasons. Notably the White British participants voiced concerns about struggling with the side-effects post-vaccination, but this did not affect their compliance and they did not express distrust in vaccination in general.

Participants, most notably a third of those from the Arab League, revealed considerable misinformation in their ethnic communities. A few African participants in particular

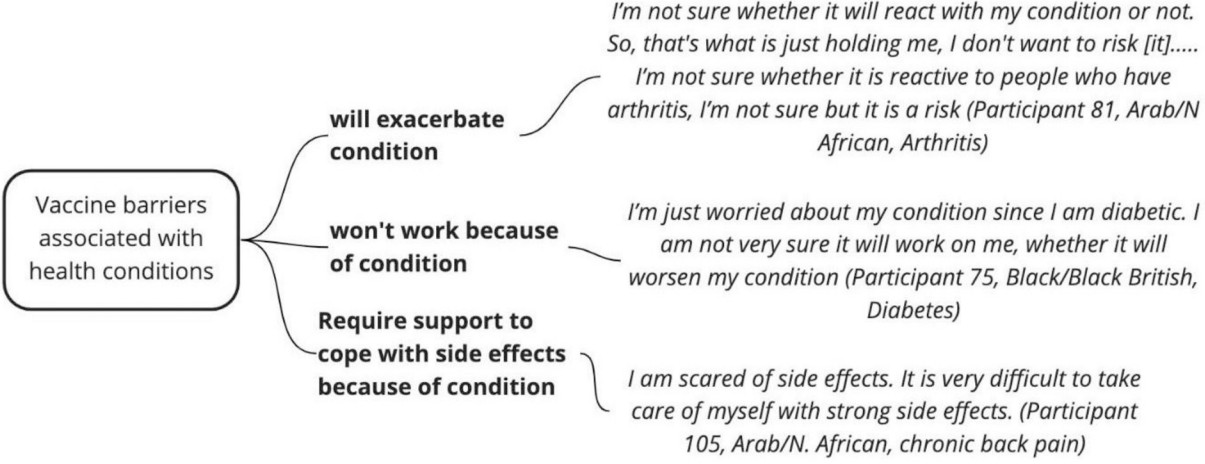

**Fig 2. Health conditions and vaccine hesitancy.**

emphasised that others in their communities spread misinformation which they did not fall for. Some feared that people born outside the UK got different vaccines or that they were being experimented on surreptitiously: "*I was a bit anxious because I didn't, obviously didn't, want to be guinea pigs. And there was talk of testing it on BAME [sic] first to see if it worked.*" (P3,SA). These fears intersected with citizenship, as undocumented migrants and asylum seekers worried that interaction with healthcare for the treatment of side effects and government programmes for vaccination itself would lead to deportation. Whilst mentioned across the core minoritised ethnic groups, misinformation was rarely suggested by Central/East European people.

Eleven participants across a range of ethnic groups and with and without disabilities, described actively researching vaccines, for example via the internet. Excepting these, and the White British and Other White participants, who took the vaccine when it was first offered, participants typically waited to see how others in the community, neighbours, family, and friends fared, before getting vaccinated themselves or being 'convinced' by them. The local Imam speaking on the radio reportedly influenced some participants to have the vaccine.

> *There was so much misinformation. . . in the Pakistani community rumours were being spread that they [vaccines] were not halal, but the Imams from the mosque did radio talks on the community channel, that there was no harm in them, and they were permissible.*

(P194,SA)

A few South Asian and North African male participants said they were the person deciding for the family. A few others from these two groups confirmed being expected to follow the decision of the head of the family, providing challenges when this conflicted with the participant's personal beliefs, as shown by the following excerpt from a young South Asian woman.

> *So, [the] first vaccine, my parents were completely 'no' and said you shouldn't have it. They were refusing to even support me to have it. So, I had to find my own way to [the centre] to receive vaccine and get back. I was frozen stiff, because I had to wait outside the vaccine centre for a taxi to return back.*

(P6,SA)

Ten Mixed Race, South Asian and Central/East European participants with various disabilities explicitly reported insufficient information from the NHS. However, it was clear from people's fears that the need for better information was far more pervasive. The mass media added to the confusion in highlighting risks factors such as blood clots. Myths and circulating misinformation were exacerbated by distrust of the Government and people resorting to internet searches.

*As science shows, vaccinations are good, so I'm pro-vaccinations but I understand that if you start to read about things people say, you can get anxious about stuff [. . .] when you read statistics and stuff like that.*

(P185,CE)

Once the decision to vaccinate was made, other factors affected its uptake. One was inconvenience, with geographical constraints intersecting with disability or economic precarity.

*[It is difficult] getting to the vaccination. I hate to use public transport. You must wait in the line for long—standing alone is suddenly to be difficult for me.*

*(P150, African, Respiratory).*

It was also difficult for a few, particularly those of our participants who were deaf, or autistic, to communicate, including the rare situation when a participant became anxious at the vaccination centre. Language issues were not specifically mentioned in this regard.

*There was a bit of a palaver because they said you don't look well. I said no I'm autistic—I'm fine I just don't want it done! [. . .] it was just that I'm not good with needles.*

*(P9, South Asian, Cognitive)*

Other individual comments were that it was hard to book online, and culturally problematic to have to expose your upper arm. Good practice was also described that mitigated some constraints:

*Because I am homebound the nurse from our practice came to our house to give the 1st and the 2nd vaccination. So, I didn't have to go to the clinic.*

*(P11,South Asian, Mobility)*

*They ensured social distancing and that everybody had their masks on. The staff was practising social distancing and keeping people in the queues. Overall, it was a good experience.*

*(P80, African, Respiratory)*

Nonetheless, participants often appeared unaware of special adjustments some centres made for disability, or their extent, and our data (see Fig 3), suggest a need for several improvements:

- support for neurodivergent individuals who found the process daunting despite a policy of streamlining them through vaccine centres.
- practical support for those unable to stand, or stand for long, especially in the queues.

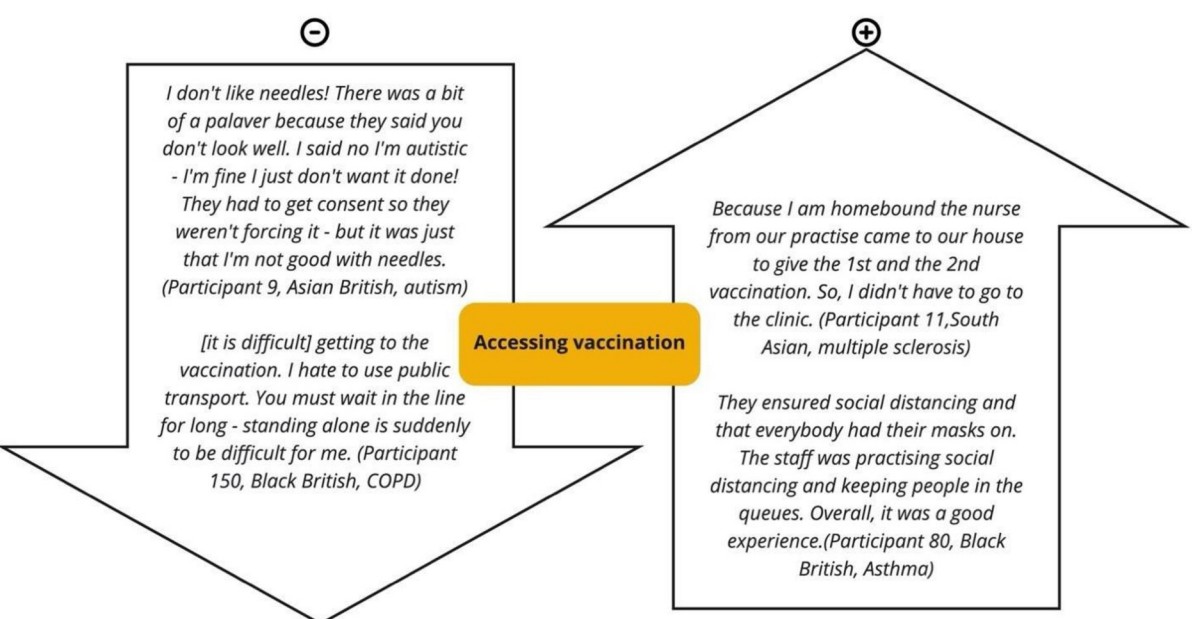

**Fig 3. Positive and negative experiences accessing vaccination with chronic conditions and disabilities.**

- more home vaccinations from nursing staff, and appointments at pharmacies and GPs when they were less busy.

- provision for informal as well as formal carers to stay with individuals.

## Freedom Day and after, July 2021 to April 2022

The UK government's gradual easing of restrictions culminated in England in the so-called 'Freedom Day' on 19 July 2021 [88]. Those who had shielded or taken great care to avoid infection for the past 18 months, especially our CEV participants, were fearful about encountering infected others including household members. This extended feelings of isolation at a time when others were recommencing social lives. A very few South Asian, Central/East European, White British and White Other participants however, like the general public, dropped all protections.

*And since the government saying let's go back to normal, it's been difficult to do that because I don't agree with it. And it was a confused kind of liberation that people were feeling. A very negative euphoria if I can say that. But we've gone with it and here we are. Everything is open and schools are back. And then you hear the cases are rising and you know that Covid hasn't gone away. It's very restrictive. It's easy for me to just stay at home. My biggest problem is my children being out of the house.*

(P207,SA)

*I just wish people would be a little bit more cognitive to other people. A little bit more sensitive to how other people feel and just take the trouble to wearing face masks. It saves potentially transmitting it. [. . .] They are on public transport, on the tubes, on the DLR, buses. And they are physically next to you. I went to see a show with my mother on Saturday [autumn 2021]*

*and the theatre at Southampton is a 3000-seater and you got the impression that it's only those over 50 that are taking notice and wearing masks. But then they're the ones that have got the most to lose because of their immune system.*

(P229,WB)

By 16 August 2021, people in England with two vaccinations were no longer legally required to self-isolate if identified as a close contact of a positive COVID-19 case. But as COVID-19 infections and hospitalisations remained high, amid concerns about the winter burden on the NHS, from 8 December 2021 mask-wearing was mandated in public places in England and working-from-home guidance and daily testing for close contacts reintroduced. It was then that we completed our first 210 interviews. Dissatisfaction with the government's handling of the pandemic continued to rise, as more and more transgressions came to light. There were calls for them to be held accountable.

*None of the politicians were following any guidelines. They didn't even wear facemasks in parliament. So, they breach one thing, one rule for them and another rule for us. So, it's a complete shambles. Yes, for families I can see like for myself, it's not practical for us to follow it to the letter but what I do agree is how come the politicians and the celebrities and, all the wealthier people, people with power, they had loopholes and things like that, and they were not punished, they all got away with it, didn't they. Matt Hancock [the then Health Secretary] and Boris [the then Prime Minister], all of them, even in parliament they weren't wearing their facemasks, all of them, they didn't have any of that on. You know, at the G8 [international intergovernmental] summit they just put their masks on for the photo op and promptly took it off after the picture was clicked. So, you know, the public are not stupid we've seen what's going on.*

(P203,SA)

By February 2022 most rules had been dropped entirely including post-infection isolation rules. Free tests ended on 1ˢᵗ April 2022. These changes led to massive staff absences in public services and transportation with almost one in three local authorities in England reporting having to ration care for elderly and disabled people [89]. Booster vaccines were offered (which several participants disengaged with) and children, returning to in-person schooling, began to be vaccinated.

In April and May 2022, we held our first series of workshops with people we had previously interviewed. There is some overlap in data with 2022 interviews; however, the longitudinal patterns are consistent across these different datasets. The second set of workshops was conducted after all interviews were completed and so is more distinct. The over-riding theme in the April workshops was of a difficult transition back to 'normal'.

People with mental health issues talked about learning to reintegrate, feeling unsettled, social pressures and the pressures of the world. Some commented that the built environment, which had changed during the pandemic, with for example new cycle lanes, changed again in 2022, making reintegration difficult without support. Participants particularly vulnerable to COVID infection reiterated the fears expressed after Freedom Day. Overall, most declared reluctance to reintegrate, continued precautionary behaviours, and in some cases avoided social situations.

*After the pandemic, I think it was Eid and we'd gone to [children's adventure park] [. . .] people were not taking precautions and it was not organised properly or anything. There were so*

*many people there, nobody had a face mask on the walk or keeping the distances. it's not safe to be out here really [. . .]. I think there's a lot of people still, in the community, you know, I'm finding, anyway, where I was, before the pandemic, quite active with my daughter, taking her out and things, and I'm sort of hesitant now. It's almost become more comfortable in a way, you know, doing this, rather than trying to do extra stuff in a way. (Bradford Workshop).*

*Sometimes if I go somewhere and if I have a mask, if I know somebody there, they'll say, "Why are you wearing a mask? You don't need to wear a mask anymore," but I feel safer in certain areas.*

(P178,SA)

## Living with COVID

The UK moved to 'living with COVID' in early 2022 [90]. In September 2022, just as our last workshops were held, an autumn COVID-19 booster programme began in England with parallel measures in other UK nations. In December 2022, the government announced the cessation of regular COVID-19 infection modelling data from January 2023, moving to monitoring COVID-19 in the same way as influenza. By April 2023 the only remnant of COVID-19 infection control measures was the spring 2023 booster programme, repeated in a more limited population (those aged over 74, living in care homes or with weakened immune systems) in autumn 2023 and again in 2024.

Most participants in our September 2022 workshops still sanitised and social distanced but mask-wearing was more variable (Fig 4). Younger participants reported that contemporaries no longer took precautions while others, feeling vulnerable to infection, said mask-wearing was now *'a lifetime thing'*.

*There's a worry that it's still there. Now everybody is interacting with each other, everyone is travelling [. . .] so you don't know, it's not something you can see on someone's face [. . .] I'm kind of worried that anyone I'm interacting with could have it, my thought is maybe it hasn't started showing. I can't just ignore that; I have to be guarded at all times (Midlands workshop)*

Overall, while still fearful of COVID-19, participants accepted the need to live with the virus. The focus had turned to the new economic crisis, and participants had largely made the transition back into society.

## Discussion and conclusions

We explored the COVID-19 containment behaviours of people with chronic conditions or impairments who are disabled in daily life and come from minoritised ethnic groups. Our findings broadly match other studies [67, 91–107] but with further nuance because of our recruitment across a range of conditions and diverse migrant groups, including undocumented and recent migrants, and our intersectional lens. For example, contrary to public and government discourses [108], we found participants tended to over-comply rather than dismiss guidelines, due to their known greater vulnerability to COVID-19. However, various structural inequities (rather than cultural values and practices [108]) could impede compliance and should be the focus of intervention. We also explored consequences of following the

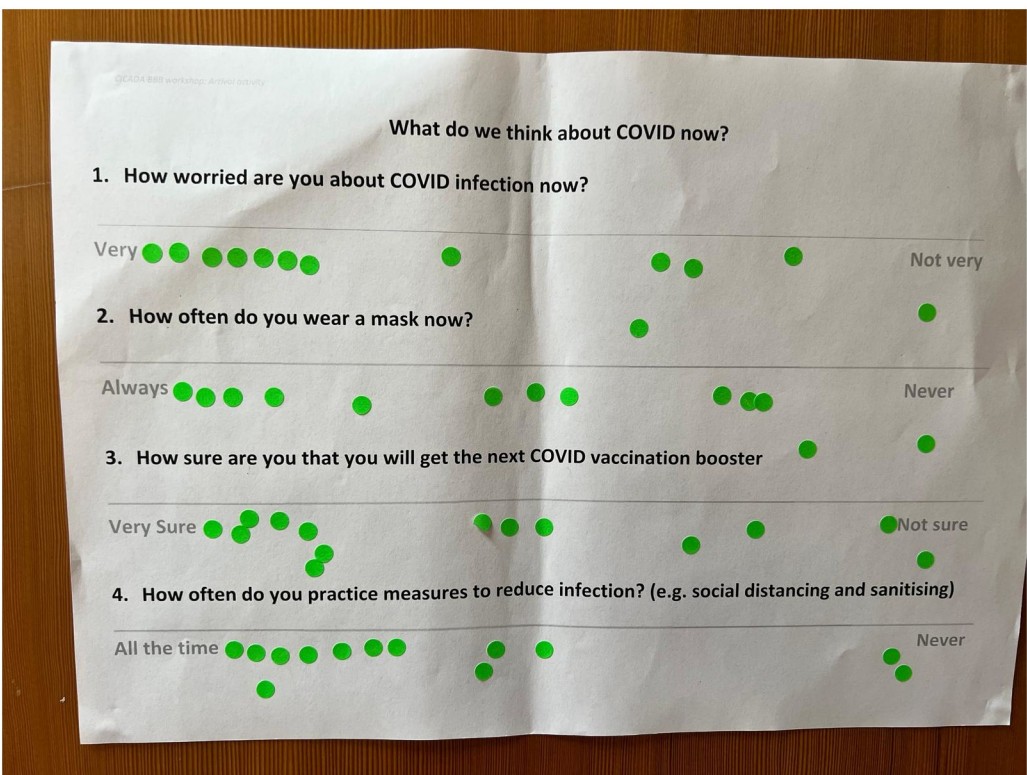

**Fig 4. Dot voting on how you feel about COVID-19 currently (Tower Hamlets, London, in person workshop).**

guidance, which, as reported for other groups, could lead to reduced mental wellbeing and sometimes create family conflict. Novel findings include the reporting by many of feeling imprisoned, and impacts on disclosures of COVID-19 infection, and related support, from the intersection with gender, citizenship, socioeconomic status and culture. Examples of problematic disclosure came from male breadwinners of North African and South Asian families, and Undocumented participants.

The need for more appropriate information targeted at minoritised ethnic groups and catering for variations in English language fluency and health literacy has been documented by others [91, 92]. Our participants added nuance in requesting clear guidance on what was safe as well as risky behaviour. They noted this was further complicated by government policies and resources being tailored to less deprived populations with more social and economic capital (see also Skovgaard-Smith [95]), hence failing to consider the barriers they faced in following the guidance. The mobility-constrained, blind, deaf, older people, and those with mental ill health, epilepsy, diabetes, multimorbidity, or lacking digital literacy or recourse to public funds felt especially neglected. The decision to reserve early shop opening times for disabled people was ill-considered in consequence [61].

Participant groups more vociferous in criticising the English government were also more likely to not comply though eventually the majority lost faith in the government. The greater compliance and confidence of the Devolved Nation participants in their government is likely to reflect the different policies applied in their regions, Wales and Scotland, and better communications.

Overall, ethnic groups at most risk from COVID-19 were most likely to comply fully with guidance; disabilities and socioeconomic status generally had more impact on non-

compliance. Nonetheless, issues complying because of a disability could contribute to the false public perception that people from ethnic minorities broke the rules, since chronic conditions such as diabetes and cardiovascular disease are disproportionately common in some minoritised ethnic groups [43]. For example, fewer than 25% of working age White British households include someone with a COVID-19-vulnerable health condition compared with nearly 33% for Pakistani and Bangladeshi households [19]. This may itself be linked to pre-existing patterns of persistent disadvantage and discrimination which worsened during the pandemic [19, 20]. Such considerations show how an intersectional approach helps avoid false and unnecessary victim-blaming, which can exacerbate racism, disableism and hate crime, as documented during the pandemic [96–100].

Many found mask-wearing and food queues at supermarkets challenging because of their health condition or impairment but some were not aware of food bank and food parcel support or wrongly believed that, if not on the CEV list, they were ineligible. Similar lack of awareness was evident regarding disability support for vaccinations. These perceptions add to our arguments that communication by the government, policymakers, and practitioners was inadequately tailored. Participants who researched formal support online felt privileged in having the knowledge and resources to do so.

Our data contribute to discussions concerning vaccination of minoritised ethnic groups. Whilst other papers have reported lower vaccination rates amongst migrants [28–31, 99], by focusing on assets and strengths we found a vaccination positivity that is rarely reported. However, ethnic minority participants needed time to deliberate and better, trusted, community-embedded information because of historical and contemporary discrimination and mistrust of the government [28–31, 99,108]. In contrast, White British participants were typically influenced by mass media, not local communities. The White groups and Central/East Europeans seemed the most empowered to research and analyse information, but the White groups were the most likely and the Central/East European participants the least likely to have been vaccinated. The intersection of health condition or disability, poverty, and living alone led to fears that vaccine side effects or interactions with existing conditions would further compromise participants' situations. Our findings about mistrust and fear of side effects correspond to those of other studies of minoritised ethnic groups [30, 109–111]. An intersection with citizenship meant undocumented migrants and asylum seekers feared deportation if using healthcare and government vaccination programmes. As with our other findings on non-compliance, ignorance of these intersectional factors can lead to victim-blaming and racism.

Participants, irrespective of citizenship or socio-economic status, disliked providing personal data for contact tracing, concerned about the destination of the data. This was reported in another pandemic study that included undocumented migrants [66], but only one other study has reported participants saying this made them prisoners in their own homes [112].

Some participants found social isolation particularly problematic in keeping them away from faith, community and social gatherings, and funerals as they came from community-facing cultures. Separation was often felt most acutely when families lived in a participant's country of origin (as also reported by Kloc-Nowak and Ryan [101]), and when migration meant a dearth of local social networks, more likely for our Central/East European than South Asian participants. However, remote connections led to greater bonding in some families which accords with a smaller study of Muslims in North-West England [91]; our undocumented migrants avoided both remote and local networking, however, to 'lay low'. It is generally acknowledged that shielding and lockdowns increased mental ill health in the general population [53, 102, 103]; we found loneliness was more common amongst those ethnic minority groups who typically do not have tight-knit diasporic communities, yet when we considered loneliness by site, it was most common in areas where there were large communities of a

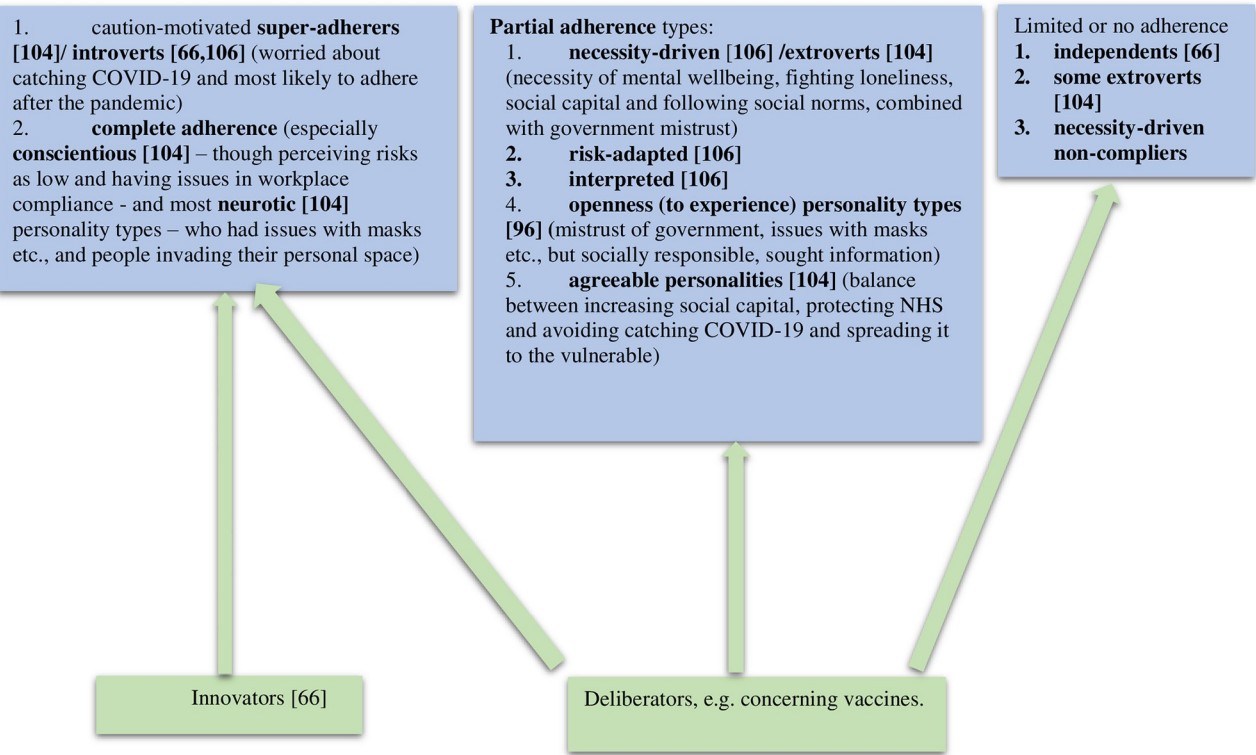

**Fig 5. Summary of the different types of compliance with COVID-19 guidance and rules.**

mixture of ethnic minority groups, a finding that needs more exploration. It might be, for example, that people from non-diasporic communities felt particularly excluded in these areas. Loneliness was more common in people with impairments that are often associated with isolation irrespective of any pandemic, such as mental health issues, or exacerbated mental health issues, such as in neurodivergent participants or those with complex care needs. By contrast, a study comparing data from the English Longitudinal Study of Ageing from 2018–19 and June-July 2020 (all respondents aged 52+) suggested a particular risk for physically disabled respondents [102]. Similar results to ours pertained in a small online cross-sectional national survey of adults in Ireland during the first lockdown [103].

We showed cultural, disability, citizenship and socio-economic nuances in the situation for those living with families. Many reported increased stress and discord in cramped or overcrowded conditions, for example with large often multigenerational families, or because of the difficulties of protecting vulnerable family members [106, 112].

Our longitudinal considerations show attitudes towards the pandemic, perceived risk and compliance with guidance changed as pandemic contexts changed. While other studies have shown compliance trajectories in the general UK population [104], many of our participants followed at least some infection-control measures for longer than the general populace, leading to anxiety and a reluctance to reintegrate because of a continued perceived vulnerability to COVID-19. However, by September 2022 participants seemed generally more concerned about the economic crisis than COVID-19.

Our findings confirm other analyses showing people cannot be simply dichotomised as "adherent" or "not-adherent" to pandemic guidance and rules. An overview of nuanced descriptors and typologies is provided in Fig 5. Perceiving their vulnerability to COVID-19,

most CICADA participants were quick to adopt infection-containment behaviours, often zealously and taking extra measures, and continued this after restrictions were lifted. Denford et al [106] interviewing 20 adults from Black African and Black Caribbean (N = 4), Asian (9) and White (N = 7) ethnic groups, called this 'caution-motivated super-adherence'. Kulpińska, Górska and Wyrwisz [66], interviewing 30 Polish migrants to the UK between October 2021 and early 2023, named as "introverts" a similar group that had maintained the pandemic way of living into 2023, being fearful of COVID-19 and having experienced mental ill health effects from the pandemic. All three studies (i.e. including CICADA) had 50–100% minoritised ethnicity representation and CICADA also a focus on disability. In CICADA, super-adherers were especially common among African, Caribbean, Mixed Race, Arab League and Undocumented participants, which may reflect reasons as diverse as fear of deportation, poor living conditions, perceived greater vulnerability and poor access to healthcare. Wright et al [104] analysed freetext data from over 17,500 respondents to the COVID-19 Social Survey, but just over 95% were White, though approximately half had long-term conditions. 'Neurotic' types were identified in Wright et al's [104] analysis as complying despite having mask issues; they feared others' risky behaviours. We found this especially amongst South Asian, White British, Arab League and Undocumented participants. Other complete adherers were named 'conscientious' types in Wright's study, more motivated by social responsibility than almost all our participants or Kulpińska, Górska and Wyrwisz' [66] introverts, who often criticised others for putting them at risk through non-compliance, and actively withdrew from others.

Kulpińska, Górska and Wyrwisz [66] distinguished another group, 'innovators', who had made many changes in their private life and personal development. We found many instances of this, which we report separately in a paper on the strengths and assets our participants used to cope with the pandemic. Here we note that in our data introverts and innovators were not mutually exclusive.

Kulpińska, Górska and Wyrwisz [66], undertaking research, like us, after the relaxing of all rules, noted a group they called 'independents', who had not followed infection-containment measures and quickly returned to normal living after the pandemic. Wright et al's [104] 'extroverts' sometimes behaved similarly, prioritising their social needs. These typologies were both rare in our data, excepting some empowered White British, Central/East European and non-disabled participants, perhaps because we considered people doubly vulnerable to COVID-19. In 2020, vulnerability was not associated with higher compliance [113], but this is likely because most people were compliant at the start. Independents were described as strongly integrated within their local community, and of relatively high socio-economic status [66]. Wright et al's [104] more highly educated responders follow the independent pattern. This agrees with our findings on empowered critiques of the guidelines and the importance of socio-economic status intersecting with other factors in shaping behaviours.

Some CICADA participants were not always able to adhere, despite wanting to. Denford et al [106] termed this "necessity-driven partial adherence", though, like Wright's extrovert partial adherers, they considered mental health and wellbeing as one necessity. Our data instead link necessity-driven partial adherence with structural inequities and ableist policies, and unlike other studies show how this compromised the wellbeing of many household members, for example causing family conflict, or the virtual imprisonment of vulnerable participants in their bedrooms.

Williams et al [105] suggested "overt rule breaking" differed from "subjective rule interpretation." In agreement, Denford et al's [106] 'risk-adapted partial adherence' group believed low risks to themselves from selective rule-breaking (e.g., not expecting to encounter others when breaking lockdown) or justified behaviours because of the uncertainty, inconsistencies, constant revision and technical language of official information on COVID-19, and lack of

clarity about local risks. We had one White British example. Denford et al's [106] 'interpreted adherence' group used similar justifications, but their behaviours were driven by their perceptions of the health and other needs of their family. Despite our participants criticising the available information they did not generally use this to justify rule-breaking except concerning vaccine hesitancy (which was not considered in the other studies); this incompletely overlaps with the rule interpreter typology, and we suggest a further group called deliberators, who needed time to make decisions. We had only four examples of agreeable types, all from South Asia (Fig 5); a few open types occurred across all ethnic groups, especially White British, Central/East European and South Asian. Some participants were totally unable to comply because of their disabilities, forming another new type that might be called 'necessity-driven non-compliers'.

Our study has limitations. Our interviews were mainly online, thus will be biased to people able to respond in this way. This may have excluded some older and more disabled participants and undocumented migrants. However, our lay co-researchers [36] principally interviewed face-to-face, some participants were from the oldest age groups, and significant numbers were undocumented. We mostly could not be explicit about which participants were Undocumented but are aware of these so were able to consider this in our general comments in the paper. We have good representation from all the targeted ethnic groups, though South Asian participants predominated. Top-up funding meant interviews continued through 2022, a challenge when analysing the data longitudinally. Memories of the early pandemic may have become less well defined in the later interviews and in all participants, recall of events before winter 2021 may have been inaccurate. However, accounts were aligned between earlier and later participants and with earlier literature. Participants may also have exhibited social desirability bias, in presenting themselves as adherent; however super-adherence was visible in face-to-face workshops, while some participants expressed strong views against COVID-19 vaccination. Interviews were conducted after Freedom Day 2021. Although participants reflected on the entire pandemic, this may have affected responses. We focused in our qualitative data on England though we have limited data from Wales and Scotland showing similar accounts and our triangulating survey (not reported here) was UK-wide. While our study was strengths-based, we have focused here on challenges; coping strategies and support are discussed in other CICADA papers. In arranging this longitudinal narrative, statements that applied throughout much of the pandemic, such as mask-wearing, and feelings of isolation and imprisonment may have been attached to single time points.

Notwithstanding, our study makes a unique contribution to the COVID-19 pandemic and inequities evidence bases. It is, we believe, the only study to have focused on the everyday experiences of people across several minoritised ethnic groups and chronic conditions and impairments. We undertook 271 interviews with members of the public and invited over half of these to two series of workshops to consider change at approximately six-monthly intervals. Unlike many other studies, we considered both compliance with guidance and rules, and the impact of complying, and through an intersectionality lens.

Overall, our findings show the importance of using an intersectional approach that highlights the impact of structural inequities, ethnocentrism and dis/ableism on compliance with infection-control measures during the COVID-19 pandemic. Our study also shows the importance of understanding the various needs of, and communicating public health messages appropriately to, groups disadvantaged by social inequities. Such considerations might shift the partial or limited adherers and deliberators to more adherent choices. Support is also needed to enable super-adherers/introverts to maintain their behaviours in the face of medical need. One size fits all messaging, policy and practice that does not account for intersectionalities and inequities is doomed to fail those most in need of support.

## Supporting information

**S1 Table. Heat maps to show the relative mention of different features of compliance by ethnicity, disability and region.** Tables a-e provide a 'big picture' quantitative content analysis of compliance by ethnic group, disability category and site, and comparison with non-core participants (the sensitivity analysis). The 'unit of analysis' is individual people. In other words, the data represent the proportion of people in each ethnic, disability, or regional group (the rows) for whom each feature (i.e. column label) was true. Percentages are rounded to two significant figures. Low percentages for a theme may simply indicate lack of response, therefore these tables are only indicative of the distribution of responses across groups. However sample sizes were sufficiently large to have some confidence in findings. Tables are Excel-generated 'heat maps', with blue shades representing low percentages and red high ones. Where percentages are low, we emphasise this in the associated paper or more commonly do not comment on that particular feature for that particular group.
(DOCX)

## Acknowledgments

We are grateful to all the participants who took part in this study, and our community co-researcher and partners who enabled or facilitated workshop events and undertook some interviews.

## Author Contributions

**Conceptualization:** Carol Rivas.

**Data curation:** Carol Rivas.

**Formal analysis:** Carol Rivas, Kusha Anand, Amanda P. Moore.

**Funding acquisition:** Carol Rivas.

**Investigation:** Kusha Anand, Amanda P. Moore.

**Methodology:** Carol Rivas.

**Project administration:** Carol Rivas, Kusha Anand, Amanda P. Moore.

**Supervision:** Carol Rivas.

**Validation:** Carol Rivas.

**Visualization:** Amanda P. Moore.

**Writing – original draft:** Carol Rivas.

**Writing – review & editing:** Kusha Anand, Amanda P. Moore.

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
