## [Decision Letter · Decision Letter 0]

8 May 2024

PONE-D-24-00940Compliance with COVID-19 government guidance and rules by disabled people and people from minoritised ethnic groups: qualitative findings from the CICADA studyPLOS ONE

Dear Dr. Rivas,

Thank you for submitting your manuscript to PLOS ONE. After careful consideration, we feel that it has merit but does not fully meet PLOS ONE’s publication criteria as it currently stands. Therefore, we invite you to submit a revised version of the manuscript that addresses the points raised during the review process.

This is an interesting paper and a valuable study, as more in-depth understanding marginalized groups' experiences during the COVID-19 pandemic is still limited. The main concern with the manuscript is that it is too long, and the description of qualitative findings needs better integration and compressing. There is also some confusion about references to quantitative data that may or may not be part of this particular set of findings. In short, the methods, findings, and discussion need some reworking to streamline and clarify information and knowledge from this study. In your revision, please make sure to respond to both reviewers' comments and update the manuscript accordingly.

We look forward to receiving your revised manuscript.

Kind regards,

Magdalena Szaflarski, PhD

Academic Editor

PLOS ONE

Journal Requirements:

2. Thank you for stating the following financial disclosure: "CR

NIHR132914

National Institute for Health and Care Research UK

https://www.nihr.ac.uk/explore-nihr/funding-programmes/health-and-social-care-delivery-research.htm

This paper presents independent research commissioned by the National Institute for Health and Care Research (NIHR). The views and opinions expressed by authors in this publication are those of the authors and do not necessarily reflect those of the NHS, the NIHR, MRC, CCF, NETSCC, the NIHR HS&DR programme or the Department of Health. The views and opinions expressed by the interviewees in this publication are those of the interviewees and do not necessarily reflect those of the authors, those of the NHS, the NIHR, MRC, CCF, NETSCC, the NIHR HS&DR programme or the Department of Health."

Reviewers' comments:

Reviewer's Responses to Questions

**Comments to the Author**

1. Is the manuscript technically sound, and do the data support the conclusions?

Reviewer #1: Yes

Reviewer #2: Yes

2. Has the statistical analysis been performed appropriately and rigorously? 

Reviewer #1: N/A

Reviewer #2: Yes

3. Have the authors made all data underlying the findings in their manuscript fully available?

Reviewer #1: Yes

Reviewer #2: Yes

4. Is the manuscript presented in an intelligible fashion and written in standard English?

Reviewer #1: Yes

Reviewer #2: Yes

5. Review Comments to the Author

Reviewer #1: This was a well written paper with a clear research purpose and interesting set of results. The findings present as a very useful historical record of the experiences of these communities during the pandemic. I thought the timeline presentation a neat way to convey the results. The discussion clearly articulated where the findings were similar to other research and the particular contributions of this research paper.

The suggested comments are minor. My main suggestion is that in places the paper could potentially be shortened to make it easier for the reader.

Tables 2 – 6 contained a lot of information that probably needed more narrative explanation so that reader could gleam the important points. It was hard to know how much these tables were contributing to the overall paper and whether some of this material was necessary.

Similarly, in places during the results there were some lengthy quotes and in some places 3 or 4 quotes where 2 would probably have been sufficient. All the material was fine and the quotes were easy to read but it seemed like the paper could have been shortened in length.

There were places during the results were there was commentary comparing quant findings between different communities. In the method section it was mentioned that a survey was conducted as part of this study but only the qualitative findings were mentioned in this paper. If there are quant findings from a survey I wasn’t sure why there was a need to turn these qualitative data into quantitative data and commentary. I assume the sample sizes were too low to conduct any inferential tests so I wasn’t sure of the value of this type of analysis. At times the numbers were helpful to orient the reader to the sample characteristics. At other times there was actual comparison of loneliness, vaccine hesitancy etc. Were their survey findings that could corroborate these findings. It was compared to previous research but this seemed out of place if it was comparing with survey findings from other quant research. This was not a major problem of the paper but maybe some attention or at least acknowledged as a limitation.

Line 107, check grammar.

Reviewer #2: The manuscript underscores the significance of incorporating viewpoints from society's most vulnerable demographics, including individuals with disabilities, racial and ethnic minorities, and undocumented individuals. I appreciated reading how the analysis and interpretation of findings integrated an intersectional approach. The methods section offered a clear and comprehensive overview of the qualitative study's methodology process.

However, due to the extensive number of interviews and collected data, the manuscript appears overly lengthy. Authors should contemplate integrating selected quotes into relevant paragraphs where they bolster major points, instead of appending them at the paragraph's end. This can help condensed manuscript. Additionally, certain themes could be merged together. Prior to resubmission, it's advisable to seek input from seasoned qualitative researchers to review the results and discussion sections thoroughly.

6. PLOS authors have the option to publish the peer review history of their article (what does this mean?). If published, this will include your full peer review and any attached files.

Reviewer #1: No

Reviewer #2: **Yes: **Jovita Murillo

---

## [Author Response · Author response to Decision Letter 0]

28 Aug 2024

Dear Editors 

I believe that we have improved the quality of the paper by taking account of the reviewers and have shortened it. The positive comments were appreciated. 

Below please find my response to the editor and journal requirement requests.

Best wishes

Carol

Editor comments 

This is an interesting paper and a valuable study, as more in-depth understanding marginalized groups' experiences during the COVID-19 pandemic is still limited. 

- Thank you

The main concern with the manuscript is that it is too long, and the description of qualitative findings needs better integration and compressing. 

- I have reduced the length by doing so – as commented on below to reviewers.

There is also some confusion about references to quantitative data that may or may not be part of this particular set of findings. 

- These data were a ‘content analysis’ of the themes. I found them helpful when ensuring accurate coverage of what was a very large dataset for thematic analysis. However, I accept that their use can be confusing and perhaps also makes for a less easy read. There were sufficient for some statistical analysis but given that sampling was purposive quota-based this would not be appropriate. It is certainly also true therefore that relativity between groups is not absolute, though there is more certainty than for smaller qualitative studies. These could perhaps therefore go in as caveated supplemental material?

The methods, findings, and discussion need some reworking to streamline and clarify information and knowledge from this study 

- I have made changes across the document to do so and this includes a couple of new references to emphasise particular aspects.

Journal Requirements 

- I have gone through these and hopefully made all the changes that are required.

2. Thank you for stating the following financial disclosure: "CR

NIHR132914

National Institute for Health and Care Research UK

https://www.nihr.ac.uk/explore-nihr/funding-programmes/health-and-social-care-delivery-research.htm

This paper presents independent research commissioned by the National Institute for Health and Care Research (NIHR). The views and opinions expressed by authors in this publication are those of the authors and do not necessarily reflect those of the NHS, the NIHR, MRC, CCF, NETSCC, the NIHR HS&DR programme or the Department of Health. The views and opinions expressed by the interviewees in this publication are those of the interviewees and do not necessarily reflect those of the authors, those of the NHS, the NIHR, MRC, CCF, NETSCC, the NIHR HS&DR programme or the Department of Health."

- Amended as here:

This paper presents independent research commissioned by the National Institute for Health and Care Research (NIHR). The views and opinions expressed by authors in this publication are those of the authors and do not necessarily reflect those of the NHS, the NIHR, MRC, CCF, NETSCC, the NIHR HS&DR programme or the Department of Health. The views and opinions expressed by the interviewees in this publication are those of the interviewees and do not necessarily reflect those of the authors, those of the NHS, the NIHR, MRC, CCF, NETSCC, the NIHR HS&DR programme or the Department of Health. The funders had no role in study design, data collection and analysis, decision to publish, or preparation of the manuscript.

- I had the below statement in the MS system which I am now also adding to the manuscript. Is this sufficient? I have added an organisational archive contact and been specific about the licence.

The full data cannot be shared publicly because they include responses from undocumented migrants who have refused this permission. The anonymised qualitative data from interviews and workshops will be deposited for archiving and reuse under a Restrictive Licence according to UCL protocols existing at the time. The Restriction is in place to protect the identities of participants who have multiple protected characteristics described in the data and to avoid anonymisation being breached. These data will be available on request for up to 25 years after end of the study (i.e. up to 2047). Archived data will be checked for anonymisation before sharing; raw data will never be shared but will remain in the UCL safe haven. Data that are considered by the custodian to be sensitive and not in the public interest will not be shared despite anonymisation. Other anonymised data will be shared according to a Restrictive Licence and extant UCL protocols. The custodian of the data to whom requests may be made is Professor Carol Rivas, c.rivas@ucl.ac.uk; requests may also be made via researchdatarepository@ucl.ac.uk.

4. Please include your full ethics statement in the ‘Methods’ section of your manuscript file. In your statement, please include the full name of the IRB or ethics committee who approved or waived your study, as well as whether or not you obtained informed written or verbal consent. If consent was waived for your study, please include this information in your statement as well The full statement has been added, and expanded to fully explain revisions to ethics.

- See p 6 last paragraph in MS

- The MS abstract was deleted and replaced with the online one.

Dear Reviewers

Thank you for your comments which I believe have improved the quality of the paper. Below please find my responses.

Reviewer 1 

This was a well written paper with a clear research purpose and interesting set of results. The findings present as a very useful historical record of the experiences of these communities during the pandemic. I thought the timeline presentation a neat way to convey the results. The discussion clearly articulated where the findings were similar to other research and the particular contributions of this research paper. 

- Many thanks for this

My main suggestion is that in places the paper could potentially be shortened to make it easier for the reader.

- This has been done as per your more specific suggestions.

Tables 2 – 6 contained a lot of information that probably needed more narrative explanation so that reader could gleam the important points. It was hard to know how much these tables were contributing to the overall paper and whether some of this material was necessary.

There were places during the results were there was commentary comparing quant findings between different communities. In the method section it was mentioned that a survey was conducted as part of this study but only the qualitative findings were mentioned in this paper. If there are quant findings from a survey I wasn’t sure why there was a need to turn these qualitative data into quantitative data and commentary. I assume the sample sizes were too low to conduct any inferential tests so I wasn’t sure of the value of this type of analysis. At times the numbers were helpful to orient the reader to the sample characteristics. At other times there was actual comparison of loneliness, vaccine hesitancy etc. Were their survey findings that could corroborate these findings. It was compared to previous research but this seemed out of place if it was comparing with survey findings from other quant research. This was not a major problem of the paper but maybe some attention or at least acknowledged as a limitation. 

- These data were only from a ‘content analysis’ of the themes. The survey data are being analysed and reported separately and do not cover all the themes considered; also just as the content analysis has proved to be, it would probably be too much for the paper. However I found the content analysis helpful when ensuring accurate coverage of what was a very large dataset for thematic analysis. Given that the tables and references to their data are confusing and perhaps also make for a less easy read, I have removed them (unless reference to the numbers was helpful). There were sufficient for some statistical analysis but given that sampling was purposive quota-based this would not be appropriate, and comparisons were made carefully so as not to suggest a greater confidence in them than was possible given the sampling approach. 

Similarly, in places during the results there were some lengthy quotes and in some places 3 or 4 quotes where 2 would probably have been sufficient. All the material was fine and the quotes were easy to read but it seemed like the paper could have been shortened in length.

- I have removed some repetitive quotes and followed a reviewer 2 comment that some could be shortened and added to the text.

Line 107, check grammar. 

- Word added for clarity

Reviewer 2: 

The manuscript underscores the significance of incorporating viewpoints from society's most vulnerable demographics, including individuals with disabilities, racial and ethnic minorities, and undocumented individuals. I appreciated reading how the analysis and interpretation of findings integrated an intersectional approach. The methods section offered a clear and comprehensive overview of the qualitative study's methodology process. 

- Many thanks

However, due to the extensive number of interviews and collected data, the manuscript appears overly lengthy. Authors should contemplate integrating selected quotes into relevant paragraphs where they bolster major points, instead of appending them at the paragraph's end. This can help condensed manuscript. Additionally, certain themes could be merged together. 

- I have taken up this suggestion to achieve some text reduction and improve some arguments, thank you. 

I was also surprised to realise that some themes were mentioned at different points as if not mentioned before, so thank you for spotting this, which has also helped reduce the manuscript. I believe the arguments are now much tighter. 

Prior to resubmission, it's advisable to seek input from seasoned qualitative researchers to review the results and discussion sections thoroughly. 

- I hope the manuscript is now satisfactory? We are all seasoned qualitative researchers, but this was a group effort leading to more repetition and poorly integrated theme work than I had realised on submission. Shortening the paper has made this easier for me to spot (!) so I hope that these errors have all been dealt with.

---

## [Editor Report · Decision Letter 1]

5 Sep 2024

Compliance with COVID-19 government guidance and rules by disabled people and people from minoritised ethnic groups: qualitative findings from the CICADA study

PONE-D-24-00940R1

Dear Dr. Rivas,

We’re pleased to inform you that your manuscript has been judged scientifically suitable for publication and will be formally accepted for publication once it meets all outstanding technical requirements.

Kind regards,

Magdalena Szaflarski, PhD

Academic Editor

PLOS ONE
---

## [Editor Report · Acceptance letter]

10 Sep 2024

PONE-D-24-00940R1 

PLOS ONE

Dear Dr. Rivas, 

I'm pleased to inform you that your manuscript has been deemed suitable for publication in PLOS ONE. Congratulations! Your manuscript is now being handed over to our production team.

Kind regards, 

on behalf of

Dr. Magdalena Szaflarski 

Academic Editor

PLOS ONE